# A Comprehensive Review of Emerging Trends and Innovative Therapies in Epilepsy Management

**DOI:** 10.3390/brainsci13091305

**Published:** 2023-09-11

**Authors:** Shampa Ghosh, Jitendra Kumar Sinha, Soumya Ghosh, Hitaishi Sharma, Rakesh Bhaskar, Kannan Badri Narayanan

**Affiliations:** 1GloNeuro, Sector 107, Vishwakarma Road, Noida 201301, India; 2ICMR—National Institute of Nutrition, Tarnaka, Hyderabad 500007, India; 3School of Chemical Engineering, Yeungnam University, 280 Daehak-Ro, Gyeongsan, Gyeongbuk 38541, Republic of Korea; 4Research Institute of Cell Culture, Yeungnam University, 280 Daehak-Ro, Gyeongsan, Gyeongbuk 38541, Republic of Korea

**Keywords:** neurostimulation, gene editing, optogenetics, treatment resistance, drug-resistant epilepsy, seizure mechanisms, cannabinoids, personalized therapy

## Abstract

Epilepsy is a complex neurological disorder affecting millions worldwide, with a substantial number of patients facing drug-resistant epilepsy. This comprehensive review explores innovative therapies for epilepsy management, focusing on their principles, clinical evidence, and potential applications. Traditional antiseizure medications (ASMs) form the cornerstone of epilepsy treatment, but their limitations necessitate alternative approaches. The review delves into cutting-edge therapies such as responsive neurostimulation (RNS), vagus nerve stimulation (VNS), and deep brain stimulation (DBS), highlighting their mechanisms of action and promising clinical outcomes. Additionally, the potential of gene therapies and optogenetics in epilepsy research is discussed, revealing groundbreaking findings that shed light on seizure mechanisms. Insights into cannabidiol (CBD) and the ketogenic diet as adjunctive therapies further broaden the spectrum of epilepsy management. Challenges in achieving seizure control with traditional therapies, including treatment resistance and individual variability, are addressed. The importance of staying updated with emerging trends in epilepsy management is emphasized, along with the hope for improved therapeutic options. Future research directions, such as combining therapies, AI applications, and non-invasive optogenetics, hold promise for personalized and effective epilepsy treatment. As the field advances, collaboration among researchers of natural and synthetic biochemistry, clinicians from different streams and various forms of medicine, and patients will drive progress toward better seizure control and a higher quality of life for individuals living with epilepsy.

## 1. Introduction

Epilepsy, a chronic neurological disorder characterized by recurrent seizures, affects millions of people worldwide [1,2,3]. Seizures result from abnormal electrical activity in the brain, leading to various physical and cognitive manifestations. While traditional antiseizure medications (ASMs) have been the cornerstone of epilepsy management for decades, a significant proportion of patients continue to experience seizures despite treatment [3]. This has spurred the exploration of innovative therapies and emerging trends in epilepsy management to address the unmet medical needs of individuals living with drug-resistant epilepsy [2]. The management of epilepsy has come a long way since its earliest descriptions in ancient texts, but challenges persist [4]. Traditional ASMs can be associated with adverse effects, including cognitive impairments, mood disturbances, and systemic toxicity [5]. Additionally, certain epilepsy syndromes may be particularly refractory to conventional treatments, necessitating alternative approaches to achieve better outcomes [6].

This review aims to comprehensively explore the latest advancements in epilepsy management, focusing on emerging trends and innovative therapies that offer new hope for individuals with drug-resistant epilepsy. By highlighting these groundbreaking approaches, we intend to shed light on potential transformative changes in the field and their implications for patient care. In the following sections, we delve into specific cutting-edge therapies and research directions that have shown promise in recent years. These include responsive neurostimulation (RNS), vagus nerve stimulation (VNS), deep brain stimulation (DBS), closed-loop stimulation, cannabidiol (CBD) as a novel adjunct therapy, the ketogenic diet, gene therapies, and the intriguing potential of optogenetics. Our primary objectives of this review are to provide a comprehensive overview of the current landscape of epilepsy management, highlighting the limitations of traditional ASMs and the need for alternative therapeutic approaches [7]. We have tried to present the latest findings and clinical evidence related to emerging therapies, including RNS, VNS, DBS, closed-loop stimulation, CBD, the ketogenic diet, gene therapies, and optogenetics. The discussions covering the assessment of the efficacy, safety, and tolerability of these innovative treatments in reducing seizure frequency and improving overall quality of life for epilepsy patients are also covered [3]. Additionally, we discuss the potential mechanisms of action underlying these therapies and their implications for understanding epilepsy pathophysiology. Finally, our objective is also to identify the potential challenges, limitations, and future research directions for each of the discussed therapies, fostering the development of more effective and patient-tailored treatments.

## 2. Traditional Approaches to Epilepsy Management

For decades, conventional ASMs have been the cornerstone of epilepsy management, providing significant relief to a large number of patients [8,9]. ASMs work by modulating the excitability of neurons, inhibiting the abnormal electrical activity that triggers seizures [10]. The advent of these medications has revolutionized epilepsy treatment and has been instrumental in achieving seizure control and improving the quality of life for many individuals with epilepsy [11]. Various classes of ASMs are available, each targeting specific mechanisms involved in seizure generation and propagation [12]. Common ASMs include phenytoin, carbamazepine, valproate, lamotrigine, and levetiracetam [13,14]. These drugs are typically prescribed based on the patient’s seizure type, epilepsy syndrome, age, and overall health. Despite their widespread use and effectiveness in a significant proportion of patients, ASMs have limitations that can hinder optimal epilepsy management. First, not all patients respond favorably to traditional ASMs, leading to drug-resistant epilepsy. Estimates suggest that approximately one-third of people with epilepsy continue to experience seizures despite adequate trials of two or more ASMs [15]. This phenomenon poses a significant clinical challenge and underscores the need for alternative therapeutic approaches to address drug-resistant epilepsy.

### 2.1. Challenges in Achieving Seizure Control with Traditional Therapies

Drug-resistant epilepsy represents a major clinical hurdle in epilepsy management [16]. Patients who are refractory to traditional ASMs face recurrent seizures that can severely impact their daily lives, disrupt social interactions, and limit educational and employment opportunities [17,18,19]. The unpredictable nature of seizures can lead to anxiety, depression, and a reduced overall quality of life. The reasons behind treatment resistance in epilepsy are complex and multifactorial. One significant challenge is the inherent variability in epilepsy. The condition is heterogeneous, and the underlying causes and mechanisms can differ greatly from one patient to another. As a result, ASMs that are effective for some individuals may not work as well for others due to differences in the brain’s structure and function [20,21]. The diversity in epilepsy subtypes, seizure types, and responses to treatment makes it challenging to achieve uniform seizure control with traditional therapies [22].

Pharmacokinetic variability is another factor contributing to treatment resistance [23]. The way ASMs are metabolized and distributed in the body can vary among individuals, affecting drug levels and therapeutic efficacy. Drug interactions and genetic factors can also influence AED metabolism, leading to differences in drug response and treatment outcomes [24,25]. This variability in drug levels can result in suboptimal seizure control and contribute to treatment resistance. Moreover, the mechanisms of action of ASMs may not address all aspects of seizure generation and propagation. While these medications primarily target ion channels and neurotransmitter receptors, certain epilepsy syndromes may involve complex networks of neurons, making them less responsive to the effects of traditional ASMs [26]. As a result, treatment with ASMs alone may not be sufficient to achieve complete seizure control in some cases [27,28]. Compliance issues also play a significant role in treatment resistance. Adherence to prescribed AED regimens is crucial for successful seizure management [29,30]. However, poor medication compliance can reduce the effectiveness of treatment and contribute to treatment resistance [29,30]. Factors such as forgetfulness, medication side effects, and the inconvenience of multiple daily doses can hinder patients’ consistent adherence to their treatment plans.

Tolerance and adaptation are additional challenges in epilepsy management [31]. Over time, some individuals may develop tolerance to the effects of ASMs, leading to decreased seizure control. The brain’s adaptability and compensatory mechanisms may reduce the long-term efficacy of certain medications, necessitating the need for alternative therapeutic approaches [32]. Furthermore, ASMs may be associated with side effects that impact treatment adherence and tolerability. Some patients may experience significant adverse effects such as dizziness, drowsiness, cognitive impairment, and mood disturbances [33]. For some individuals, these side effects may outweigh the benefits of seizure reduction, leading to treatment discontinuation or non-compliance.

Addressing the challenges of drug-resistant epilepsy requires a comprehensive and individualized approach. As we delve into the world of emerging therapies, it is important to recognize that traditional ASMs continue to be vital in managing epilepsy for many patients [34]. However, the limitations of these therapies underscore the need for innovative and personalized treatments. By understanding the complexities of treatment resistance and identifying novel targets, such as specific genes or neural circuits, researchers can develop more effective therapies to improve seizure control and enhance the quality of life for individuals living with epilepsy. As we move forward, collaboration between researchers, clinicians, and patients will play a pivotal role in advancing the field of epilepsy management, driving us closer to the day when drug-resistant epilepsy becomes a challenge of the past [35]. The pursuit of emerging trends and innovative therapies, along with a deeper understanding of the underlying mechanisms of epilepsy, offers hope for improved therapeutic options and a brighter future for people living with epilepsy. Through continued research, dedication, and unwavering commitment, we can transform the lives of those affected by epilepsy and pave the way for more effective and personalized treatments.

### 2.2. The Need for Novel Treatment Approaches to Address Drug-Resistant Epilepsy

The persistence of drug-resistant epilepsy highlights the critical necessity for novel and innovative treatment approaches [22,36,37]. Research and clinical efforts have intensified in recent years to develop therapies that target specific epilepsy subtypes, identify novel drug targets, and explore non-pharmacological interventions. The emergence of new technologies and a deeper understanding of the underlying mechanisms of epilepsy have paved the way for innovative therapeutic strategies [22]. Neurostimulation devices, such as RNS, VNS, and DBS, offer potential alternatives for patients who are unresponsive to traditional ASMs (Figure 1) [37]. Additionally, advancements in precision medicine and personalized approaches hold promise for tailoring treatments to individual patients based on their unique genetic and molecular profiles [38,39] Gene therapies are also being explored as potential treatments for certain genetic epilepsy syndromes [40].

Non-pharmacological interventions such as the ketogenic diet and CBD have gained attention because of their anticonvulsant properties and have shown benefits in reducing seizure frequency in some patients [6,41]. The limitations of traditional therapies underscore the urgency to explore and implement innovative treatments to improve outcomes for individuals living with epilepsy [42].

## 3. Responsive Neurostimulation

Responsive neurostimulation (RNS) is a cutting-edge therapy designed to provide personalized and adaptive treatment for drug-resistant epilepsy [43,44]. The RNS system consists of a small neurostimulator device that is surgically implanted in the skull, along with one or two intracranial electrodes placed in or near the epileptogenic brain region responsible for seizure initiation [45,46,47]. The system operates on the fundamental principle of closed-loop neurostimulation, wherein it continuously monitors brain activity and delivers electrical stimulation in response to detected abnormal patterns [48]. The mechanism of action of RNS includes the following: (1) Monitoring brain activity: The implanted electrodes continuously record the electrical signals from the brain, detecting subtle changes that precede the onset of a seizure. Advanced algorithms within the RNS system analyze the recorded brain activity in real time. (2) Detecting seizure onset: The RNS system is programmed to recognize specific patterns of electrical activity associated with the onset of a seizure. This personalized detection algorithm is tailored to each individual based on their unique seizure characteristics. (3) Responsive stimulation: Once the system detects the pre-defined seizure activity, it delivers brief electrical pulses to the epileptic brain region. The stimulation is intended to disrupt the abnormal neural firing patterns and prevent the seizure from fully manifesting. (4) Adaptation and learning: The RNS system is designed to adapt and learn over time. As it continuously monitors brain activity and stimulation effectiveness, it can refine its algorithms to optimize seizure detection and stimulation parameters for each patient, enhancing treatment efficacy [46,48].

The efficacy of RNS in reducing seizure frequency and improving the quality of life for individuals with drug-resistant epilepsy has been supported by clinical studies and real-world evidence [49,50] In a pivotal clinical trial, the RNS system demonstrated significant benefits for patients with medically refractory focal epilepsy [51,52]. The trial included participants who experienced an average of eight or more disabling partial-onset seizures per month despite treatment with multiple ASMs. Results showed that RNS-treated patients experienced a substantial reduction in seizure frequency, with a median seizure reduction of 44% at one year and 53% at two years after implantation [51]. Furthermore, long-term follow-up studies and real-world experiences have reinforced the positive outcomes observed in the initial clinical trial [44]. Real-world evidence has shown that RNS provides sustained and durable seizure reduction, leading to improved seizure control and enhanced quality of life for patients over extended periods [52,53]. Additionally, RNS has demonstrated particular effectiveness in patients with seizures arising from focal brain regions that are not amenable to resective surgery, making it a valuable treatment option for those who are not suitable candidates for other surgical interventions [48,54].

### Potential Side Effects and Safety Considerations in RNS

As with any medical intervention, RNS is associated with potential side effects and safety considerations [55]. However, it is crucial to recognize that adverse events associated with RNS are generally manageable and often outweighed by the benefits of seizure reduction [56]. The surgical risks included in the surgical procedure to implant the RNS system are the typical risks associated with brain surgery, such as infection, bleeding, and anesthesia-related complications [57]. However, advances in neurosurgical techniques have minimized the risk of these complications. There may be some stimulation-related side effects of RNS. For example, some patients may experience mild side effects related to the electrical stimulation, such as tingling sensations, muscle twitches, or changes in mood or cognition [58]. These effects are generally temporary and tend to diminish over time as the brain adapts to the stimulation. Nevertheless, there is also a possibility of hardware-related issues during the process of RNS [59,60]. The RNS system is a sophisticated medical device that requires regular monitoring and maintenance. Battery replacements and system adjustments may be necessary over time, and patients should remain under the care of a specialized epilepsy team [61].

Studies show that RNS might have some cognitive and memory effects [57]. While RNS is designed to minimize cognitive side effects, some individuals may experience mild cognitive changes, particularly during the early stages of treatment [62]. These effects are often localized to the brain region being stimulated and tend to be reversible upon adjustment of stimulation parameters. Nonetheless, RNS represents a promising and innovative therapeutic approach for drug-resistant epilepsy. By providing adaptive and personalized treatment based on real-time brain activity, RNS offers the potential to significantly reduce seizure frequency and improve the quality of life for patients who have not responded to traditional ASMs [57,63,64]. Although there are potential side effects and safety considerations associated with the treatment, the benefits of improved seizure control and enhanced quality of life make RNS a valuable addition to the armamentarium of epilepsy management options [57,65,66,67]. Nevertheless, it is essential to understand that RNS, along with deep brain and closed-loop stimulation, requires intracranial EEG monitoring and surgical device implantation, which inherently restricts its application to a specific subset of patients. This limitation arises from the meticulous assessment and individualized implantation procedures necessitated by the invasive nature of these interventions. While acknowledging this constraint, it is important to highlight that the potential benefits offered by RNS, particularly for patients who have exhausted alternative treatment avenues, underscore the ongoing need for research and technological advancements to make this approach more accessible and broaden its impact in the realm of epilepsy management.

## 4. Vagus Nerve Stimulation

Vagus nerve stimulation (VNS) is a neuromodulation therapy that involves the implantation of a device that stimulates the vagus nerve, a major nerve that extends from the brainstem to various organs in the body, including the heart and digestive system [68,69,70]. The VNS system consists of a small generator, typically implanted under the skin in the chest, connected to a lead wire that is wrapped around the left vagus nerve in the neck [70]. The exact mechanism by which VNS exerts its anticonvulsant effects is not fully understood, but it is believed to involve several interconnected processes. The vagus nerve plays a crucial role in the regulation of various bodily functions, and its stimulation is thought to modulate the balance of neuronal activity in the brain, promoting inhibitory pathways and dampening excessive excitatory activity that can lead to seizures [71,72].

VNS is designed to provide continuous, intermittent electrical stimulation to the vagus nerve at pre-defined parameters. This stimulation has been shown to reduce seizure frequency and severity in patients with drug-resistant epilepsy [73,74]. By altering the activity of brain regions involved in seizure generation, VNS helps to prevent the spread of abnormal electrical activity and disrupts the development of seizures [75]. VNS is primarily used as an adjunctive therapy for patients with partial-onset seizures that have not responded well to traditional ASMs [75,76]. Clinical studies have demonstrated that VNS can lead to a significant reduction in seizure frequency, with some patients experiencing a 50% or greater reduction in seizure occurrence [77]. Moreover, the benefits of VNS tend to increase over time, with long-term treatment associated with further improvements in seizure control [78,79].

### 4.1. Recent Advancements in VNS Technology

In recent years, advancements in VNS technology have focused on enhancing treatment efficacy and patient convenience. One notable improvement is the development of closed-loop or on-demand VNS systems, also known as responsive VNS [80,81]. These systems utilize real-time EEG monitoring to detect seizure activity and deliver VNS stimulation automatically when abnormal brain activity is detected [81]. By targeting stimulation specifically during seizure events, responsive VNS aims to optimize therapy effectiveness while minimizing side effects [81,82]. Furthermore, advancements in device design and programming options have allowed for more personalized and precise stimulation parameters. Clinicians can now tailor the VNS settings to individual patients, adjusting stimulation parameters such as pulse width, frequency, and intensity to optimize the therapeutic response [79,80,82]. This customization enables a more patient-centric approach, which may lead to improved seizure control and tolerability. Additionally, rechargeable VNS devices are being introduced, eliminating the need for regular battery replacement surgeries. These devices can be recharged externally, making the treatment more convenient for patients and reducing the burden of frequent surgical procedures [83,84].

### 4.2. Ongoing Research and Clinical Trials in VNS for Various Epilepsy Syndromes

VNS continues to be an area of active research, with ongoing clinical trials investigating its potential benefits for various epilepsy syndromes and patient populations. Studies are exploring the safety and efficacy of VNS in children with drug-resistant epilepsy. Early intervention with VNS may offer advantages in preventing cognitive and developmental delays associated with uncontrolled seizures in pediatric patient [85,86]. Lennox–Gastaut syndrome (LGS) is a severe and treatment-resistant childhood epilepsy syndrome [87]. Clinical trials are evaluating the effectiveness of VNS in reducing drop attacks and other seizure types characteristic of LGS [88]. Ongoing research and clinical trials are also exploring the applicability of VNS in other epilepsy syndromes, with the aim of expanding the range of patients who can benefit from this innovative therapy. Nevertheless, research is ongoing to identify subgroups of patients with drug-resistant focal epilepsy who may benefit the most from VNS. This includes investigating potential biomarkers and predictive factors for VNS responsiveness [89]. Novel VNS approaches, such as non-invasive vagus nerve stimulation (nVNS), are being studied to explore their effectiveness as a less invasive alternative to traditional VNS therapy [90,91]. Some studies are investigating the potential synergistic effects of combining VNS with other neuromodulation techniques or with specific ASMs to enhance seizure control [92,93,94]. As VNS research continues to evolve, ongoing clinical trials hold the promise of further elucidating the therapeutic potential of VNS in various epilepsy syndromes and refining patient selection criteria for optimal outcomes. Therefore, the understanding to date is that VNS has emerged as a valuable adjunctive therapy for drug-resistant epilepsy [95]. The stimulation of the vagus nerve through the implantation of a VNS device leads to neuromodulation, resulting in the modulation of neural activity to reduce seizure frequency and improve overall seizure control. Recent advancements in VNS technology, including responsive or closed-loop systems and customizable stimulation parameters, offer the potential for improved treatment efficacy and patient outcomes.

It is imperative to recognize that VNS, while holding promising therapeutic potential, is not devoid of adverse effects, akin to many medical interventions. As said above, the mechanism of VNS involves the modulation of neural pathways via electrical impulses to the vagus nerve, with the intent of influencing neuronal activity and, consequently, ameliorating epileptic seizures. However, the intricacy of neural interactions can result in unintended repercussions. Stimulation of vagus nerve afferent fibers, responsible for conveying sensory information from peripheral tissues to the CNS, can induce various adverse outcomes. One notable consequence is vocal cord dysfunction, wherein the vagus nerve’s aberrant stimulation can disrupt the coordinated movements of the vocal cords during respiration, potentially leading to hoarseness, stridor, or even difficulty breathing [96]. Moreover, the stimulation may provoke laryngeal spasms, triggering involuntary contractions of the laryngeal muscles and further exacerbating respiratory difficulties. Concurrently, the activated vagus nerve fibers can provoke a cough reflex, causing persistent or severe coughing episodes that can be distressing and hinder daily functioning [97,98]. Additionally, the VNS-induced afferent signaling can elicit dyspnea, characterized by subjective feelings of breathlessness or discomfort during breathing [96,98]. This respiratory distress can be particularly concerning for individuals with compromised lung function. Moreover, the stimulation may evoke sensations of nausea and vomiting, which can detrimentally impact an individual’s overall well-being and compliance with the treatment regimen [97]. Furthermore, an intriguing but intricate association emerges between VNS and sleep apnea [99,100]. The electrical impulses targeting the vagus nerve’s afferent fibers can inadvertently influence respiratory centers in the brainstem, potentially altering breathing patterns during sleep [99,101]. This disruption is evidenced by an elevation in the apnea-hypopnea index, indicative of increased instances of sleep apnea events characterized by pauses in breathing or shallow breathing during slumber. The complex interplay between neural regulation, vagal stimulation, and respiratory control underscores the need for vigilant monitoring and individualized approaches when implementing VNS as an adjunctive therapy for epilepsy. By comprehensively addressing the intricate web of neural interactions and physiological consequences, we provide a more holistic perspective for clinicians, researchers, and individuals considering VNS as part of their therapeutic strategy.

## 5. Deep Brain Stimulation

Deep brain stimulation (DBS) is an advanced neuromodulation technique that has shown promise in the management of epilepsy, particularly in individuals with drug-resistant seizures [102,103]. Originally developed as a treatment for movement disorders such as Parkinson’s disease, DBS has evolved as a potential therapeutic option for patients whose epilepsy remains uncontrolled despite medical and surgical interventions [104]. Unlike traditional open-loop neurostimulation, DBS is designed to deliver electrical impulses to specific brain regions in a controlled and targeted manner, aiming to modulate aberrant neural activity associated with seizure generation [105,106]. DBS for epilepsy typically targets specific brain structures that are known to be involved in the initiation and propagation of seizures.

The most common target for DBS in epilepsy is the anterior nucleus of the thalamus (ANT), a region involved in relaying sensory and motor signals to the cerebral cortex [107]. The rationale behind targeting the ANT is based on its role in the limbic system, which plays a significant role in regulating emotions and behaviors, including seizure activity. The delivery of electrical stimulation to the ANT aims to alter the network dynamics of the limbic system, effectively dampening the excessive excitability that can lead to seizure development [108,109]. By disrupting the synchronization of neuronal firing patterns, DBS helps prevent the spread of abnormal electrical activity throughout the brain, reducing the likelihood of seizures [110]. Additionally, some studies have explored alternative targets, such as the hippocampus and subthalamic nucleus, with promising results in specific patient populations [111,112,113].

### Clinical Evidence Illustrating the Efficacy and Safety of DBS

DBS holds promise as a promising neuromodulation technique for the treatment of drug-resistant epilepsy. By targeting specific brain regions involved in seizure generation, DBS aims to modulate neural activity and disrupt the propagation of abnormal electrical patterns. Numerous clinical studies and case reports have provided evidence for the efficacy and safety of DBS in reducing seizure frequency and improving overall seizure control in drug-resistant epilepsy [48,114,115]. A landmark multicenter randomized controlled trial (NCT00101933), known as the SANTE (Stimulation of the Anterior Nucleus of the Thalamus in Epilepsy) trial, demonstrated the effectiveness of DBS in reducing seizures [116]. The study involved individuals with drug-resistant epilepsy who received either active stimulation or sham stimulation. The results showed a significant reduction in seizure frequency in the active stimulation group, with >41% of patients at 1-year follow-up and >68% of patients at in 5-years follow-up experiencing a 50% or greater reduction in seizure frequency. Moreover, long-term follow-up studies of the SANTE trial participants have shown sustained benefits of DBS over time, with continued reductions in seizure frequency and improvements in quality of life observed years after the initial implantation [116]. By the end of the first year and continuing through the fifth year, both the Liverpool Seizure Severity Scale and the 31-item Quality of Life in Epilepsy measure demonstrated substantial improvements over their respective baselines. These improvements were statistically significant [116].

Case studies and real-world experiences have also contributed to the growing body of evidence supporting DBS efficacy [117,118]. Many of these reports involve patients with various types of epilepsy, including those with different etiologies and seizure semiologies [117,119,120]. These case studies have consistently demonstrated the positive impact of DBS on seizure control and have provided valuable insights into patient selection criteria, optimal stimulation parameters, and the potential risks and benefits of the procedures [120]. Regarding safety, DBS has generally been well-tolerated in the majority of patients. Adverse effects related to the stimulation itself are typically mild and transient, such as tingling sensations or muscle contractions [121]. Serious complications are infrequent but can include infection, lead migration, or hardware-related issues [122]. Overall, the risks associated with DBS need to be carefully balanced against the potential benefits, particularly in individuals with severe and drug-resistant epilepsy.

## 6. Closed-Loop Stimulation

Closed-loop stimulation, also known as on-demand or responsive stimulation, is an innovative neurostimulation approach that represents a significant advancement over traditional open-loop neurostimulation [123,124,125,126]. While open-loop neurostimulation involves delivering electrical impulses at pre-defined intervals or continuous patterns, closed-loop systems dynamically adjust stimulation based on real-time feedback from the patient’s brain activity [125,126,127]. This real-time feedback is typically obtained through the continuous monitoring of brain signals, such as via electroencephalography (EEG) or electrocorticography (ECoG), which provide valuable information about the brain’s electrical activity [128]. The primary advantage of closed-loop stimulation is its ability to adapt to the patient’s physiological state and dynamically intervene at the earliest signs of abnormal brain activity, such as pre-seizure or prodromal patterns [129]. By detecting these patterns in real time, closed-loop systems can promptly deliver targeted stimulation precisely when and where it is needed, effectively preventing the progression of seizures before they fully manifest [125,130]. This personalized approach not only improves the efficacy of the neurostimulation but also minimizes the risk of overstimulation and potential side effects that can occur with continuous, open-loop stimulation [131,132].

### 6.1. Recent Studies and Trials Evaluating Closed-Loop Systems

Recent studies and clinical trials investigating closed-loop systems have shown promising results in preventing seizures and improving seizure control in patients with drug-resistant epilepsy. One notable closed-loop stimulation device that has been evaluated in clinical trials is the NeuroPace RNS System [133,134]. The clinical trial for this system demonstrated significant seizure reduction in patients with medically refractory focal epilepsy [135,136]. Results showed that patients experienced a median reduction of 70% in seizure frequency at 12–15 months after implantation. Additionally, a subset of patients achieved a remarkably greater reduction in seizures, highlighting the potential for substantial seizure control with closed-loop stimulation [136]. Moreover, closed-loop systems have demonstrated the ability to detect and respond to specific brain patterns associated with seizures, allowing for the optimization of stimulation parameters and personalized treatment [131,134]. Some studies have shown that closed-loop stimulation can be tailored to individual patients, resulting in improved efficacy compared to standard open-loop approaches [137,138,139,140].

### 6.2. Potential for Personalized Closed-Loop Approaches

The potential for personalized closed-loop approaches in epilepsy management is a particularly exciting area of research. Each individual’s epilepsy is unique, with variations in seizure types, triggers, and brain activity patterns. Closed-loop systems have the inherent capability to capture and analyze this individual variability, allowing for the development of personalized treatment strategies [141,142,143]. Harnessing the power of patient-specific data, including EEG or ECoG recordings, genetic profiles, and clinical history, closed-loop systems have the potential to tailor the timing, intensity, and stimulation site to align precisely with each patient’s unique seizure patterns and needs. For example, a closed-loop system can be programmed to recognize the early signs of seizure activity in a particular patient and deliver stimulation precisely at those critical moments to halt seizure progression [144]. Furthermore, closed-loop systems have the potential to learn and adapt over time, continuously refining their algorithms and responsiveness based on patient feedback and long-term outcomes [105]. As technology and data analytics advance, closed-loop approaches are expected to become increasingly sophisticated, leading to further improvements in seizure prediction and prevention [120].

Despite the promising potential of closed-loop stimulation, challenges remain in optimizing closed-loop algorithms, validating their reliability, and determining the most effective stimulation parameters for different patient populations [120,131]. Additionally, the implementation of closed-loop systems requires robust data processing capabilities, advanced algorithms, and accurate seizure prediction models [124,143]. However, closed-loop stimulation represents a significant advancement in neurostimulation therapies for epilepsy management. By dynamically responding to real-time brain activity, closed-loop systems offer personalized and adaptive treatment approaches that can effectively prevent seizures before they fully develop.

## 7. Cannabidiol and Epilepsy

Cannabidiol (CBD) is one of the many compounds found in the *Cannabis sativa* plant, commonly known as hemp or marijuana. Unlike tetrahydrocannabinol (THC), another well-known cannabinoid, CBD does not produce psychoactive effects and is not associated with the feeling of being “high” [145,146]. CBD has gained significant attention in recent years for its potential therapeutic properties, including its anticonvulsant effects [146]. The exact mechanisms of action by which CBD exerts its anticonvulsant effects are not fully understood, but several potential pathways have been proposed. CBD is thought to interact with the endocannabinoid system, a complex signaling network that regulates various physiological processes in the body, including neural excitability [147,148]. By modulating the endocannabinoid system, CBD may reduce excessive neuronal excitability, which is a key factor in seizure generation [149,150]. CBD is also believed to influence other non-cannabinoid receptor systems, such as serotonin and transient receptor potential (TRP) channels, which play roles in pain perception, mood regulation, and neuroprotection [151]. These diverse interactions contribute to the multifaceted mechanisms of CBD in epilepsy management.

### 7.1. Clinical Trials and Evidence Supporting the Use of CBD

The anticonvulsant properties of CBD have been extensively studied (Figure 2), leading to the approval of a CBD-based medication for certain epilepsy syndromes. Epidiolex® (cannabidiol) oral solution, a pharmaceutical-grade CBD formulation, has been approved by regulatory agencies for the treatment of specific epileptic conditions [152]. One of the most compelling lines of evidence supporting the use of CBD is its effectiveness in reducing seizures in patients with Dravet syndrome and Lennox–Gastaut syndrome (LGS), two severe childhood epilepsy syndromes that are notoriously challenging to manage with conventional therapies. Clinical trials of Epidiolex® in Dravet syndrome and LGS have shown a significant reduction in seizure frequency compared to placebo, leading to the approval of this medication for these specific indications [153]. Recent evidence has also provided valuable insights into the use of CBD for other epilepsy types and syndromes. Observational studies and patient registries have reported favorable responses to CBD in reducing seizure frequency and improving seizure control in various pediatric and adult epilepsy populations [5]. However, it is essential to note that individual responses to CBD can vary, and not all patients experience the same degree of benefit.

### 7.2. Concerns and Considerations Regarding the Use of CBD

While CBD shows promise as an adjunctive therapy for epilepsy, there are several concerns and considerations that warrant attention. CBD can interact with certain medications metabolized by the liver’s cytochrome P450 enzyme system, potentially affecting their effectiveness or safety [154,155]. Therefore, it is essential for patients and healthcare providers to be aware of potential drug interactions when using CBD alongside other medications [5,156]. CBD is generally well-tolerated, but some individuals may experience side effects, such as fatigue, diarrhea, and changes in appetite or weight. Most side effects are mild and transient, but patients should be closely monitored during CBD treatment [156].

The regulatory landscape surrounding CBD products varies by country and region [157,158]. In some areas, CBD may be available as a prescription medication, while in others, it may be available as an over-the-counter supplement. It is crucial for patients to use high-quality CBD products from reputable sources to ensure safety and efficacy [159]. Not all patients respond to CBD in the same way, and some may not experience significant seizure reduction. It is essential to set realistic expectations and monitor the patient’s response to CBD therapy over time [159,160]. As with any new medication, the long-term safety of CBD requires further investigation, especially in populations that may require prolonged or continuous use. Finding the optimal dose of CBD for each patient can be challenging, and a gradual titration may be necessary to achieve the best therapeutic effect [161]. CBD has emerged as a potential adjunctive therapy for certain epilepsy syndromes, particularly Dravet syndrome and Lennox–Gastaut syndrome [162,163]. Its mechanisms of action are thought to involve interactions with the endocannabinoid system and other receptor systems in the brain [164]. Clinical trials and other evidence have demonstrated the efficacy of CBD in reducing seizure frequency and improving seizure control in specific epilepsy populations [165]. However, concerns and considerations, such as potential drug interactions with CBD, side effects, and individual response variations, highlight the need for careful patient selection, monitoring, and further research for its usage.

### 7.3. Emerging Therapeutic Avenues

In addition to cannabidiol, several other new antiepileptic drugs (ASMs) have emerged as potential therapeutic options for epilepsy management. One such notable candidate is cenobamate, which has shown promising results in the treatment of refractory focal epilepsy [166,167]. Cenobamate’s broad efficacy profile and the potential for achieving seizure freedom in challenging cases have garnered attention [167]. Clinical trials have demonstrated significant reductions in seizure frequency and notable improvements in seizure control [167,168]. With its novel mechanism of action targeting voltage-gated sodium channels, cenobamate presents a unique approach to addressing drug-resistant epilepsy [169]. While further studies are required to fully elucidate its long-term safety and efficacy, cenobamate holds promise as a valuable addition to the armamentarium of ASMs for epilepsy. The evolving landscape of epilepsy treatment highlights the ongoing efforts to provide patients with a diverse range of effective therapeutic options, each tailored to address specific needs and challenges. As new drugs such as cenobamate continue to demonstrate their potential, research and innovation remain pivotal in enhancing the quality of life for individuals living with epilepsy.

## 8. Ketogenic Diet and Epilepsy

The ketogenic diet is a high-fat, low-carbohydrate, and moderate-protein dietary approach designed to mimic the metabolic state of fasting [10,28]. When adhering to a ketogenic diet, the body shifts from primarily using carbohydrates for energy to utilizing fats as its primary fuel source [170]. This metabolic shift leads to the production of ketone bodies in the liver, which can provide an alternative energy source for the brain [171]. The therapeutic use of the ketogenic diet for epilepsy management dates back to the early 1920s when it was first introduced as a potential treatment for drug-resistant epilepsy [9,172,173]. The diet was initially implemented as a means to mimic the beneficial effects of fasting, which was known to reduce seizure frequency in some individuals. Over time, researchers and clinicians refined the diet’s composition and established specific ratios of fat, carbohydrates, and protein to optimize its effectiveness while maintaining nutritional balance [172,174].

Current research has continued to explore the efficacy of the ketogenic diet as a valuable and non-pharmacological treatment option for epilepsy [172]. Numerous clinical studies and trials have investigated the diet’s impact on seizure control in both pediatric and adult populations with various forms of drug-resistant epilepsy [175,176]. A meta-analysis of multiple studies revealed that approximately 50% of patients on the ketogenic diet experienced a significant reduction in seizure frequency, with around 10–15% achieving complete seizure freedom [177]. While the diet’s response may vary among individuals, evidence consistently indicates that the ketogenic diet can lead to clinically meaningful seizure reduction in a substantial proportion of patients [177,178]. Moreover, recent studies have expanded the applications of the ketogenic diet beyond refractory epilepsy, exploring its potential benefits in other neurological conditions, including some neurodevelopmental disorders and brain-tumor-related epilepsy [179].

### Potential Mechanisms of Action and Variations of the Ketogenic Diet

The exact mechanisms by which the ketogenic diet exerts its anticonvulsant effects are not entirely understood, but several hypotheses have been proposed [180,181]. One of the key factors contributing to the diet’s efficacy is the elevation of ketone bodies in the bloodstream, which is believed to have anticonvulsant properties [182,183]. Ketone bodies provide an alternative fuel source for the brain, supporting neuronal function and stabilizing excitability, potentially reducing the likelihood of seizures [180,183]. Moreover, the ketogenic diet may influence the balance of neurotransmitters in the brain [183,184]. By promoting a reduction in excitatory neurotransmitters and an increase in inhibitory neurotransmitters, the diet can help dampen seizure activity and contribute to better seizure control [185].

Moreover, the diet’s impact on energy metabolism is thought to play a significant role in its anticonvulsant effects. By altering energy metabolism in the brain, the ketogenic diet may affect the availability of adenosine triphosphate (ATP), the primary energy currency of cells [178,186]. This alteration in energy availability can impact neuronal activity and contribute to seizure suppression [184]. The classic ketogenic diet typically consists of a 4:1 ratio of fats to combined carbohydrates and protein, with approximately 90% of the daily caloric intake coming from fats. Carbohydrates and protein are significantly restricted in this approach [187]. However, to accommodate different patient populations and preferences, variations of the ketogenic diet are available. These include the modified Atkins diet and the low glycemic index treatment (LGIT), which have lower fat-to-carbohydrate and protein ratios and are often easier to implement and sustain [188]. These variations can be particularly appealing for adolescents and adults seeking the benefits of the ketogenic diet without the strict adherence to the classic 4:1 ratio.

The ketogenic diet can be tailored to suit individual patient needs. Factors such as age, underlying medical conditions, dietary preferences, and lifestyle can all be considered in customizing the diet. This personalized approach may enhance adherence and increase the likelihood of successful seizure control [178,187]. Hence, the ketogenic diet remains a valuable and well-established therapeutic option for epilepsy management, particularly for drug-resistant epilepsy. Its historical role in epilepsy treatment has been reinforced by contemporary research, which continues to demonstrate its efficacy in reducing seizure frequency and improving seizure control. While the exact mechanisms of action are still under investigation, the diet’s ability to induce ketosis and alter brain metabolism likely plays a pivotal role in its anticonvulsant effects. The ketogenic diet stands as a testament to the significant impact that dietary interventions can have in the field of epilepsy [178,189]. For individuals seeking alternative or complementary treatments for their condition, the ketogenic diet may offer new possibilities and hope for improved seizure management and a better quality of life.

## 9. Gene Therapies for Epilepsy

Gene therapy is an innovative approach aimed at treating diseases by modifying or manipulating the genetic material of cells [190,191]. The fundamental principle of gene therapy is to correct or replace faulty genes that contribute to the development or progression of a particular condition. In the context of epilepsy, gene therapy holds the potential to address the underlying genetic abnormalities that give rise to seizure disorders [192]. By targeting specific genes associated with epilepsy, gene therapy aims to restore normal cellular function and inhibit seizure generation, providing a promising avenue for the development of novel and more targeted epilepsy treatments [40,186,193].

Recent advancements in gene editing technologies have revolutionized the field of gene therapy, enabling more precise and efficient targeting of specific genes. One of the most revolutionary gene editing tools is CRISPR-Cas9, which allows scientists to edit DNA sequences with remarkable accuracy [194]. With CRISPR-Cas9, researchers can modify or delete epilepsy-related genes and investigate their impact on seizure susceptibility [194]. For epilepsy, gene editing techniques are being utilized to explore the role of various genes implicated in the disorder. By targeting genes associated with ion channels, neurotransmitter receptors, or cellular signaling pathways, researchers can investigate how alterations in these genes contribute to epileptogenesis [195,196]. Additionally, gene editing tools are being used to correct disease-causing mutations in patient-derived cells or animal models, potentially paving the way for personalized gene therapies tailored to specific genetic defects [197,198,199,200].

### Safety and Efficacy of Gene Therapies

While gene therapy for epilepsy is still in the early stages of development, preclinical studies in animal models have shown promising results [184,201,202]. Animal models with specific epilepsy-related genetic mutations have been treated using gene therapy approaches, resulting in reduced seizure frequency and improved seizure control [202,203,204,205]. The preclinical studies have provided valuable insights into the potential therapeutic benefits of gene therapies and have identified potential target genes for further investigation [205]. In terms of clinical studies, several gene therapy trials for epilepsy are currently underway or in the planning stages. These trials aim to evaluate the safety and efficacy of gene therapies in human patients with specific genetic forms of epilepsy. One notable example is the development of adeno-associated virus (AAV) vectors as a delivery system for gene therapies [206]. AAV vectors have shown promise as a means to deliver therapeutic genes to specific brain regions in a controlled and targeted manner [206,207]. It is important to note that gene therapy approaches for epilepsy face unique challenges. The complexity of the brain and the diversity of genetic factors contributing to epilepsy necessitate rigorous evaluation of potential risks and benefits. Delivery methods, such as viral vectors, must be carefully engineered to ensure accurate and efficient gene transfer while minimizing immune responses and other adverse effects [207,208].

Long-term safety and potential off-target effects of gene editing in the brain remain areas of active investigation and consideration [209]. Ethical considerations, such as ensuring informed consent and addressing concerns about permanent genetic modifications, are crucial in developing responsible gene therapies for epilepsy [191]. Nevertheless, gene therapy represents a promising and innovative frontier in epilepsy treatment [210]. By targeting specific genes associated with epileptogenesis, gene therapies hold the potential to provide more precise and personalized treatments for individuals with genetic forms of epilepsy [211]. While challenges remain, gene therapies have the potential to revolutionize epilepsy treatment and improve the lives of individuals living with this challenging neurological disorder.

Emerging advancements in gene therapy hold significant promise for revolutionizing the landscape of epilepsy treatment. Gene therapy offers a groundbreaking approach to address the genetic anomalies contributing to seizure disorders by manipulating cellular genetic material. Through gene editing technologies such as CRISPR-Cas9, researchers can now target specific epilepsy-associated genes with remarkable accuracy, shedding light on the intricate molecular mechanisms underlying epileptogenesis. Recent preclinical studies in animal models demonstrate promising results, with gene therapy interventions effectively reducing seizure frequency and enhancing seizure control. Clinical trials focusing on human patients with specific genetic forms of epilepsy are underway, evaluating the safety and efficacy of gene therapies. While challenges in delivering genes to the brain and ensuring long-term safety remain, gene therapy’s potential to provide precise, personalized treatments for genetic forms of epilepsy is groundbreaking. Ethical considerations and rigorous evaluation are imperative, but the prospects of gene therapies offer hope for revolutionizing epilepsy treatment and improving the quality of life for those impacted by this complex neurological disorder.

## 10. Optogenetics in Epilepsy Research

Optogenetics is a cutting-edge technique that combines genetics and optics to control the activity of specific neurons in living tissue using light. This revolutionary method involves genetically engineering neurons to express light-sensitive proteins called opsins, which respond to specific wavelengths of light [212,213]. When these opsins are activated by light, they can either stimulate or inhibit the activity of the targeted neurons. Optogenetics allows precise and real-time manipulation of neural circuits, providing researchers with unprecedented control to study the function of specific brain regions and the mechanisms underlying various neurological disorders, including epilepsy [212,213,214].

In epilepsy research, optogenetics plays a vital role in unraveling the complex neural dynamics that lead to seizure generation and propagation [215]. By selectively activating or silencing specific populations of neurons in animal models of epilepsy, researchers can investigate the causal relationships between neural activity patterns and seizure development [205,216]. Optogenetics provides a powerful tool to explore how abnormal neuronal firing, circuit interactions, and network synchronization contribute to epileptic phenomena, thus advancing our understanding of epilepsy pathophysiology. Optogenetics has yielded groundbreaking findings in epilepsy research, shedding light on the intricate neural processes underlying seizure activity [217,218]. In animal models of epilepsy, researchers have utilized optogenetics to selectively activate or inhibit specific neuron populations in brain regions implicated in seizure generation [219,220]. One significant discovery is the identification of “seizure hotspots” in the brain, regions with a higher propensity to initiate and propagate seizures. By optogenetically stimulating these seizure hotspots, researchers have been able to trigger epileptic activity and observe the patterns of seizure propagation in real time. Conversely, inhibiting these regions using optogenetics can prevent seizure development, highlighting the critical role of these brain areas in epileptogenesis [219]. Additionally, optogenetics has elucidated the role of specific cell types, such as inhibitory interneurons, in controlling network excitability and seizure susceptibility [221]. By targeting these interneurons with optogenetic tools, researchers have shown that modulating their activity can either enhance or suppress seizure activity, providing insights into potential therapeutic strategies [222,223]. Optogenetics has been instrumental in investigating the dynamics of brain circuitry during seizures. By simultaneously recording and manipulating neuronal activity using optogenetics, researchers have gained a deeper understanding of how seizures spread through interconnected brain regions and how specific circuit abnormalities contribute to epileptic events.

### Future Possibilities of Optogenetics in Clinical Applications

The potential clinical applications of optogenetics in epilepsy are both promising and challenging [215,216]. While optogenetics has primarily been used in preclinical research, its translation to clinical practice faces several significant hurdles. Directly applying optogenetics to human brains is currently not feasible due to the need for gene delivery and light-delivery systems that would require invasive procedures. However, the knowledge gained from optogenetics experiments in animal models can inform the development of more targeted and effective therapies. Optogenetics research can help identify specific neural targets or circuit components that can be manipulated through alternative means, such as targeted pharmacological interventions or neuromodulation techniques [224].

Optogenetics research can inspire the development of novel closed-loop neurostimulation systems [225]. By combining optogenetic techniques with responsive neurostimulation technologies, it may be possible to create closed-loop systems that detect aberrant neural activity and deliver precisely timed light stimulation to prevent or disrupt seizure activity [130,226]. This hybrid approach could potentially offer a personalized and adaptive therapy for individuals with drug-resistant epilepsy. Looking further ahead, advances in non-invasive techniques for optogenetic activation and the development of novel light-sensitive proteins may make it possible to non-invasively apply optogenetics to human brains [227,228,229]. Although this goal is still in the realm of basic research, it holds promise for the future of epilepsy therapy and other neurological disorders. Optogenetics is a powerful tool in epilepsy research, enabling precise control of neural activity to study the mechanisms underlying seizures. By combining optogenetics with other neurostimulation approaches, such as responsive neurostimulation, the future possibilities for personalized and adaptive epilepsy treatments are indeed promising [142,230].

## 11. Discussions

In this comprehensive review, we explored a range of innovative therapies for epilepsy management. We began with an overview of traditional approaches to epilepsy treatment, highlighting the limitations of conventional ASMs and the need for novel treatment strategies to address drug-resistant epilepsy. Importantly, poor medication compliance in ASMs can reduce the effectiveness of treatment in the long term and contribute to treatment resistance. It is widely recognized that consistent and timely adherence to prescribed medication regimens is crucial for achieving optimal outcomes in epilepsy management. When patients do not follow their medication schedule as prescribed, the therapeutic levels of antiseizure medications in their bloodstream may become insufficient, leading to breakthrough seizures and reduced seizure control. Over time, this can result in a reduced response to the medication, making the condition more resistant to treatment and necessitating adjustments to the treatment plan. We then delved into three emerging therapies: RNS, VNS, and DBS, discussing their mechanisms of action, clinical evidence, and potential benefits for different epilepsy syndromes. Next, we examined the role of CBD in epilepsy treatment, emphasizing the pharmacology of CBD, clinical trials supporting its use in specific epilepsy syndromes, and considerations for its adjunctive therapy. We further explored the potential of gene therapies and optogenetics in epilepsy research. Gene therapies offer a promising path to target specific genes associated with epilepsy, with recent advancements in gene editing techniques showing potential for precise and personalized treatments. Optogenetics, on the other hand, has enabled groundbreaking findings in understanding seizure mechanisms by allowing real-time manipulation of neural circuits in animal models. Although direct clinical applications of optogenetics in humans are currently challenging, the knowledge gained from this research may inform the development of future therapies.

The landscape of epilepsy management is rapidly evolving, with emerging trends and novel therapies continually reshaping the field. Staying updated with the latest research findings and advancements is critical for healthcare providers, researchers, and patients alike. Awareness of innovative therapies, such as responsive neurostimulation, gene therapies, and optogenetics, allows for informed decision making and the exploration of new treatment options for patients with drug-resistant epilepsy. Moreover, ongoing research and clinical trials may lead to the approval of new treatments and expanded indications for existing therapies. As epilepsy is a complex and heterogeneous disorder, understanding the full spectrum of available treatment options ensures that patients receive personalized and effective care tailored to their specific needs.

The review of emerging therapies for epilepsy management has provided insights into promising directions for future research. First and foremost, continued investigation into the mechanisms of action of RNS, VNS, DBS, and gene therapies is essential to optimize their therapeutic benefits and refine patient selection criteria. Long-term safety and efficacy data from clinical trials are crucial to ensure the responsible and evidence-based use of these therapies in routine practice. Furthermore, exploring the potential of combining various therapeutic approaches may yield synergistic benefits for epilepsy treatment. For instance, combining pharmacological treatments with neurostimulation techniques or using optogenetics to study how different therapies interact with specific neural circuits could lead to more comprehensive and tailored treatment regimens. The application of artificial intelligence (AI) and machine learning algorithms to epilepsy research and treatment is another promising avenue. AI-based algorithms can analyze large datasets, such as EEG recordings or genetic information, to identify patterns associated with seizure risk and treatment response [231,232]. Integrating AI with closed-loop neurostimulation systems may enable real-time seizure prediction and personalized neuromodulation, enhancing seizure control and quality of life for patients [233,234,235,236]. Additionally, ongoing research in the field of CBD and other cannabinoids may uncover new therapeutic applications and optimize dosing regimens for different epilepsy syndromes. Continued clinical trials will provide critical evidence for the long-term safety and efficacy of CBD and its potential role as a monotherapy or adjunctive treatment. Finally, collaborative efforts between researchers, clinicians, and industry partners are vital for advancing epilepsy management. Increased investment in research and development will drive the translation of preclinical findings into innovative therapies and ultimately benefit individuals living with epilepsy.

## 12. Conclusions

In the last few years, there has been significant progress made in epilepsy management through innovative therapies. From responsive neurostimulation and neuromodulation techniques, such as VNS and DBS, to the exploration of CBD as an anticonvulsant agent and the cutting-edge fields of gene therapies and optogenetics, researchers have expanded the horizons of epilepsy treatment. The importance of staying updated with emerging trends cannot be overstated, as new discoveries may offer hope for patients with drug-resistant epilepsy and open avenues for more targeted and effective therapies. Future research should focus on refining existing therapies, exploring combination approaches, and harnessing AI and machine learning to optimize epilepsy management. Collaborative efforts among researchers, clinicians, and industry partners will be key to realizing the full potential of these innovative therapies and advancing epilepsy care to new frontiers. Ultimately, the goal of ongoing research and progress in epilepsy management is to improve the lives of individuals living with epilepsy, providing them with greater seizure control, improved quality of life, and renewed hope for a brighter future.

## Figures and Tables

**Figure 1 brainsci-13-01305-f001:**
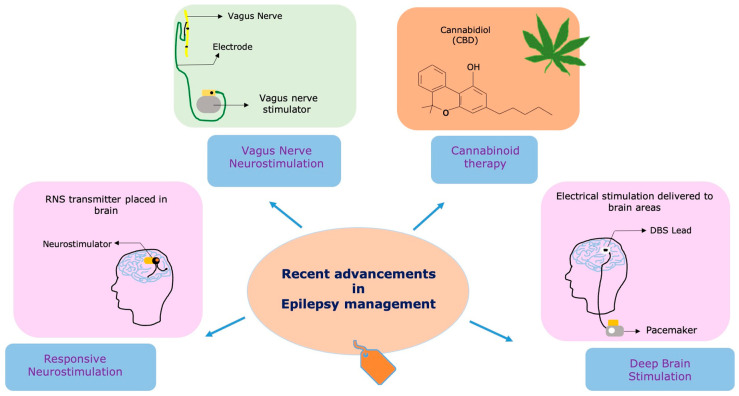
Recent technologies and therapies used in the management of epilepsy. The figure illustrates cutting-edge treatments and innovative therapies utilized in epilepsy management. Responsive neurostimulation (RNS)—An implantable neurostimulator device that detects and responds to abnormal brain activity, providing on-demand electrical stimulation to reduce seizure frequency. Vagus nerve stimulation (VNS)—A device that stimulates the vagus nerve through electrical impulses, helping to modulate brain activity and decrease seizure occurrences. Deep brain stimulation (DBS)—A neuromodulation technique that involves implanting electrodes in specific brain regions to deliver electrical stimulation and regulate neural activity for seizure control. Cannabidiol (CBD)—A natural compound derived from the cannabis plant, known for its potential anticonvulsant effects and often used as an adjunct therapy for certain epilepsy syndromes.

**Figure 2 brainsci-13-01305-f002:**
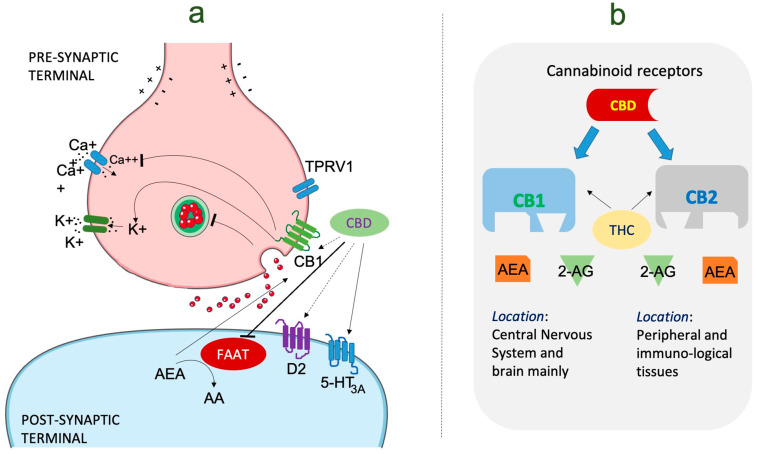
The mechanism of action of cannabidiol in the management of epilepsy. (**a**) Cannabinoid molecular mechanism of action in epilepsy. CBD performs an inhibitory role on FAAT resulting in activation of CB1, CB2, and TRPV1 receptors. Anandamide levels are also increased due to FAAH inhibition. (**b**) This image represents the type of cannabinoid receptors present in the human endocannabinoid system, i.e., CB1 and CB2, found in varying parts of body. It is a system having a lock and key mechanism with specific functions performed by both CB1 and CB2. CB1 has greater affinity for both THC and AEA as compared to CB2. Abbreviations: CBD: cannabinoid; AEA: anandamide; CB1: cannabinoid receptor1; D2: dopamine receptor 2; FAAH: fatty acid amide hydrolase. TRPV1: transient receptor potential vanilloid 1, THC: tetrahydrocannabinol.

## Data Availability

Available on request.

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
