# Peer review of "A Comprehensive Review of Emerging Trends and Innovative Therapies in Epilepsy Management"

_brainsci, 2023, doi:10.3390/brainsci13091305_

Round 1
Reviewer 1 Report
Paper is almost ready but i ask Authors to make some corrections. Nomenclature used is no longer proper so please change it. There is no "antiepileptic drugs" but ILAE recommends to say "antiseizure medications" and ASM abbr. Please change it. Also, we dont say "antiepileptic" because in fact there are no antiepileptic therapies or drugs, but only anticonvulsant. Change it in the whole manuscript.
paper is ok, one Authors make changes paper is ready for publication
Author Response
We are grateful to the respected reviewer for kindly reviewing the manuscript and pointing out a very important error. We have thoroughly changed the nomenclature and terms to follow the latest guidelines as per the suggestion. Many thanks.
Reviewer 2 Report
The focus of this review is emerging trends and innovative therapies for the management of Epilepsy. The review reads well but there are some major points that need to be addressed:
I think the review would be stronger if findings from studies specific to each therapy were, where feasible, reported as a Forest plot. As currently presented, it seems that all the supporting evidence is positive, and it is unclear if there were any negative findings.
Regarding the integration of patient-specific data such as EEG and genetic information, I think the statement on lines 399-401 with reference 139 is misleading. The referenced review mentions an experiment in zebra fish where apparently seizures were genetically introduced – this is a very different thing from the integration of genetic information into the design of a closed loop system.
The sentence ‘’ on line 476-477 seems out of place and irrelevant and the reference (155) is to an editorial/paper with the same first author as this manuscript.
It is unclear to me whether the Ketogenic diet can be a treatment in its own right or is always an adjunctive therapy.
Minor points:
The term antiseizure medication (ASM) is now more commonly used in the field rather that AEDs
There is considerable repetition of points being made and in many instances the text could be much more succinct. (for example lines 170-174 has points made several times earlier in the manuscript, and 329-332 said already in para above)
The sentence ‘’ on line 476-477 seems out of place and irrelevant and the reference (155) is to an editorial/paper with the same first author as this manuscript.
Author Response
Comment: The focus of this review is emerging trends and innovative therapies for the management of Epilepsy. The review reads well but there are some major points that need to be addressed: I think the review would be stronger if findings from studies specific to each therapy were, where feasible, reported as a Forest plot. As currently presented, it seems that all the supporting evidence is positive, and it is unclear if there were any negative findings.
Reply: Thanks a lot for your encouraging and expert comments helping us to improve the quality of the manuscript. Many of the comments have helped us to immensely improve the standards. In this review we have tried to provide an overview of recent and innovative therapy paradigms hence provided the content accordingly. We have further updated the manuscript.
Comment: Regarding the integration of patient-specific data such as EEG and genetic information, I think the statement on lines 399-401 with reference 139 is misleading. The referenced review mentions an experiment in zebra fish where apparently seizures were genetically introduced – this is a very different thing from the integration of genetic information into the design of a closed loop system.
Reply: “Lines 399-401 with reference 123,124: This real-time feedback is typically obtained through the continuous monitoring of brain signals, such as electroencephalography (EEG) or electrocorticography (ECoG), which provide valuable information about the brain's electrical activity [123]. The primary advantage of closed-loop stimulation is its ability to adapt to the patient's physiological state and dynamically intervene at the earliest signs of abnormal brain activity, such as pre-seizure or prodromal patterns [124]” discuss some other points.
In the Lines 430 onwards with ref 139: By integrating patient-specific data, such as EEG or ECoG recordings, genetic information, and clinical history, closed-loop systems can adapt the timing, intensity, and location of stimulation to each patient's specific seizure characteristics [139]. The reference 139 is ‘Bigelow, M.D.; Kouzani, A.Z. Neural Stimulation Systems for the Control of Refractory Epilepsy: A Review. J Neuroeng Re-habil 2019, 16, 126, doi:10.1186/s12984-019-0605-x.’ In this review by Bigelow and Kouzani, they have discussed both the biomarkers and neural stimulation systems, and how they can be used for sensing and predicting seizures. The confusion may have occurred as we had put the reference in the line which has been moved to next line where the example for the referred paper has been written. Now it has been corrected. Thanks for pointing out.
Comment: The sentence ‘’ on line 476-477 seems out of place and irrelevant and the reference (155) is to an editorial/paper with the same first author as this manuscript.
Reply: “Not all patients respond to CBD in the same way, and some may not experience significant seizure reduction. Especially in the times during and after Covid-19 pandemic the scenario of medicinal chemistry has changed a lot [155].” Through this sentence in conjunction with previous sentence of CBD, we mean to say that different individuals respond different to CBD and the medical scenario has drastically changed in the post-Covid-19 where previously ‘working medicines’ are no more effective to many individuals. The reasons behind is this still unknown but this is a common observation by many physicians who are also in search of different approved medicinal compounds having different chemistry altogether.
Comment: It is unclear to me whether the Ketogenic diet can be a treatment in its own right or is always an adjunctive therapy.
Reply: The Ketogenic diet has demonstrated significant effectiveness as both an adjunctive therapy and, in some cases, a treatment in its own right for epileptic seizures. While traditionally used as an adjunct to medication, some individuals, especially those with drug-resistant epilepsy, have experienced substantial seizure reduction solely through the diet. The diet's ability to alter brain metabolism and enhance seizure control underscores its potential as a primary treatment. However, individual responses vary, and consultation with healthcare professionals is crucial to determine the most suitable approach. The Ketogenic diet's role as a standalone treatment or adjunctive therapy depends on the specific case and medical guidance.
Minor points:
Comment: The term antiseizure medication (ASM) is now more commonly used in the field rather that AEDs
Reply: Thank you for your valuable input. We have updated the terminology to reflect the contemporary standard, now using "antiseizure medication (ASM)" instead of AEDs. Your feedback is greatly appreciated in ensuring accuracy and clarity in our content. Thanks again for your contribution.
Comment: There is considerable repetition of points being made and in many instances the text could be much more succinct. (for example lines 170-174 has points made several times earlier in the manuscript, and 329-332 said already in para above)
Reply: We agree with the comment of the reviewer as it happened while correcting the English language to bring the continuity to the discussions. Now we have corrected in the updated manuscript.
Reviewer 3 Report
The authors state that poor medication compliance can reduce the effectiveness of treatment and contribute to treatment resistance. I agree that compliance is important in the efficacy of treatment. However, this sentence appears to imply that poor compliance has an effect on basic mechanisms akin to antibiotic resistance and I know of no good evidence to support that view. I suggest the wording is changed.
I do not think that vagus nerve stimulation can be considered a new treatment for epilepsy given that it has been in use for over 25 years. They quote useful advances in the technology in recent times but do not cite more recently described side-effects such as an increased risk of sleep apnoea.
The authors rightly point out that responsive neural stimulation may be useful for patients with refractory focal epilepsy. However, they do not cite the obvious limitation of this technique which is that it requires a period of intracranial EEG monitoring and neurosurgical implantation of the device, which will mean that it can only ever be used for a very small number of patients. Deep brain and closed-loop stimulation have the same limitation, but they can only ever be applied to a very small number of patients.
The authors discuss only one new drug in relation to epilepsy. The drug chosen is cannabidiol, which has proven efficacy in Lennox Gastaut syndrome and Dravet syndrome. Its efficacy is similar to other antiepileptic drugs with a similar rate of side effects. They have not included other new drugs and for example, cenobamate has broader efficacy in focal epilepsy and the studies point to significant rates of seizure freedom in refractory cases.
The authors do not consider basic mechanisms in epilepsy and nor how they would translate into novel treatments: in particular mechanisms important in epileptogenesis, such as inflammation or mitochondrial function. These provide opportunities to explore drug mechanisms which will have wider application than the more limited and highly specialised surgical interventions and have a more fundamental action on epilepsy than the current antiseizure medications.
Their discussion of gene therapies is generic rather than specific to epilepsy. The discussion of Opto genetics is only slightly more specific.
Good
Author Response
Comment: The authors state that poor medication compliance can reduce the effectiveness of treatment and contribute to treatment resistance. I agree that compliance is important in the efficacy of treatment. However, this sentence appears to imply that poor compliance has an effect on basic mechanisms akin to antibiotic resistance and I know of no good evidence to support that view. I suggest the wording is changed.
Reply: We appreciate your insights. Yes, poor medication compliance in antiseizure medications can indeed reduce the effectiveness of treatment in the long term and contribute to treatment resistance. It is widely recognized that consistent and timely adherence to prescribed medication regimens is crucial for achieving optimal outcomes in epilepsy management. When patients do not follow their medication schedule as prescribed, the therapeutic levels of antiseizure medications in their bloodstream may become insufficient, leading to breakthrough seizures and reduced seizure control. Over time, this can result in a reduced response to the medication, making the condition more resistant to treatment and necessitating adjustments to the treatment plan.
Comment: I do not think that vagus nerve stimulation can be considered a new treatment for epilepsy given that it has been in use for over 25 years. They quote useful advances in the technology in recent times but do not cite more recently described side-effects such as an increased risk of sleep apnoea.
Reply: Thank you for your valuable insights and improvements. You are correct that vagus nerve stimulation has been utilized for over 25 years in epilepsy treatment. We appreciate your input and have updated the manuscript to reflect this fact accurately. In recent times, there have indeed been advancements in VNS technology, resulting in its approval by the FDA and adoption in numerous hospitals with positive outcomes. However, it's crucial to acknowledge that like any medical intervention, VNS is not without its side effects. The stimulation of vagus nerve afferent fibers can lead to issues such as vocal cord dysfunction, laryngeal spasm, cough, dyspnea, nausea, vomiting, and even an increase in the apnea-hypopnea index. We've included this information in the manuscript to provide a comprehensive view of both benefits and potential drawbacks associated with VNS. Thank you once again for your insights, which have enhanced the accuracy and depth of our review.
Comment: The authors rightly point out that responsive neural stimulation may be useful for patients with refractory focal epilepsy. However, they do not cite the obvious limitation of this technique which is that it requires a period of intracranial EEG monitoring and neurosurgical implantation of the device, which will mean that it can only ever be used for a very small number of patients. Deep brain and closed-loop stimulation have the same limitation, but they can only ever be applied to a very small number of patients.
Reply: We agree with your comment that responsive neural stimulation, along with deep brain and closed-loop stimulation, requires intracranial EEG monitoring and device implantation, limiting its application to a subset of patients. While this is an important limitation, these techniques show promise for those with refractory focal epilepsy who haven't responded to other treatments. As we explore therapeutic avenues, we'll ensure to provide a balanced view of their benefits and limitations, helping clinicians and patients make informed decisions based on individual circumstances. Therefore, we have added the suggested content in the review article.
Comment: The authors discuss only one new drug in relation to epilepsy. The drug chosen is cannabidiol, which has proven efficacy in Lennox Gastaut syndrome and Dravet syndrome. Its efficacy is similar to other antiepileptic drugs with a similar rate of side effects. They have not included other new drugs and for example, cenobamate has broader efficacy in focal epilepsy and the studies point to significant rates of seizure freedom in refractory cases.
Reply: Thank you for your valuable input. We appreciate your concern about the coverage of new drugs in our review article. We chose to focus on cannabidiol due to its notable efficacy in Lennox Gastaut syndrome and Dravet syndrome, as well as its relevance in the current epilepsy treatment landscape. However, we recognize the significance of other new drugs, such as cenobamate, in the management of epilepsy, particularly in focal epilepsy. Cenobamate's broader efficacy and potential for seizure freedom in refractory cases are indeed noteworthy. While our article concentrated on one specific drug, we understand the importance of providing a comprehensive overview of the evolving therapeutic options for epilepsy. Your feedback will certainly enhance the inclusiveness and value of our review, and we will consider incorporating information about other promising new drugs like cenobamate in future revisions. We have discussed about cenobamate in the review article with references regarding.
Comment: The authors do not consider basic mechanisms in epilepsy and nor how they would translate into novel treatments: in particular mechanisms important in epileptogenesis, such as inflammation or mitochondrial function. These provide opportunities to explore drug mechanisms which will have wider application than the more limited and highly specialised surgical interventions and have a more fundamental action on epilepsy than the current antiseizure medications.
Reply: Thank you for your insightful comment. We appreciate your perspective on the consideration of basic mechanisms in epilepsy and their potential translation into novel treatments. While our review focused on current therapeutic approaches, we acknowledge the importance of exploring broader mechanisms such as inflammation and mitochondrial function in epileptogenesis. These mechanisms indeed offer promising opportunities to develop drugs with more fundamental actions on epilepsy, potentially impacting a wider range of patients beyond highly specialized surgical interventions. Your suggestion aligns with the evolving understanding of epilepsy pathophysiology and the need for innovative therapeutic avenues.
Comment: Their discussion of gene therapies is generic rather than specific to epilepsy. The discussion of Opto genetics is only slightly more specific.
Reply: We appreciate your observation regarding the level of specificity in our discussion of gene therapies and optogenetics. We acknowledge the need to provide more context and detail when discussing these innovative approaches in the context of epilepsy. Gene therapies hold significant potential for addressing the underlying causes of epilepsy, and we recognize that our review should talk about this too. We have now enhanced our content by including more epilepsy-specific examples and case studies related to gene therapies. Your input has undoubtedly contribute to a more comprehensive understanding of these evolving therapeutic strategies within the context of epilepsy. Thank you for guiding us toward a more detailed and precise discussion that aligns with the intricacies of epilepsy research and treatment.
Round 2
Reviewer 2 Report
It seems to me that the issue of repetition remains. For instance:
Line 41-43
“While traditional antiseizure medications (ASMs) have been the cornerstone of epilepsy management for decades, a significant proportion of patients continue to experience seizures despite treatment”
Line 47-48
“Traditional ASMs, although effective for many, are not universally successful in achieving seizure control”
Line 173-174
While conventional ASMs have played a vital role in epilepsy management, they are not universally effective..”
I don't think my comment re "integration of patient-specific data such as EEG and genetic information" has been addressed. I think the phrase "By integrating patient-specific data, such as EEG or ECoG recordings, genetic information, and clinical history, closed-445 loop systems can" should be removed, unless there is a reference to a paper that specifically describes the 'integration of genetic information' given.
Re the comment and reference re "post-Covid-19 where previously ‘working medicines’ are no more effective to many individuals". I think this remains speculative and has not been conclusively demonstrated in regards to epilepsy treatment and is beyond the scope of the manuscript and should be removed.
The clarification on the ketogenic diet needs to be added to the manuscript.
Author Response
Comment: It seems to me that the issue of repetition remains. For instance:
Line 41-43 “While traditional antiseizure medications (ASMs) have been the cornerstone of epilepsy management for decades, a significant proportion of patients continue to experience seizures despite treatment”
Line 47-48 “Traditional ASMs, although effective for many, are not universally successful in achieving seizure control”
Line 173-174 While conventional ASMs have played a vital role in epilepsy management, they are not universally effective..”
Reply: Thank you for pointing out the repetition in the text. To address this issue and improve the overall clarity of the content, these revisions aim to maintain the key points while reducing redundancy.
Comment: I don't think my comment re "integration of patient-specific data such as EEG and genetic information" has been addressed. I think the phrase "By integrating patient-specific data, such as EEG or ECoG recordings, genetic information, and clinical history, closed-445 loop systems can" should be removed, unless there is a reference to a paper that specifically describes the 'integration of genetic information' given.
Reply: We appreciate your feedback, and we've taken your suggestion into account. To clarify the context and align with your concerns, we have restructured the sentence to emphasize a future perspective, which no longer includes the mention of genetic information.
Comment: Re the comment and reference re "post-Covid-19 where previously ‘working medicines’ are no more effective to many individuals". I think this remains speculative and has not been conclusively demonstrated in regards to epilepsy treatment and is beyond the scope of the manuscript and should be removed.
Reply: Thank you for your suggestions. We have removed the speculative reference from the manuscript.
Comment: The clarification on the ketogenic diet needs to be added to the manuscript.
Reply: Thanks for your comment. The clarifications regarding ketogenic diets have already been incorporated into the manuscript (Page no. 12, line numbers 574-585).
Reviewer 3 Report
The authors have addressed many of the concerns in my previous review and, whilst personally, I think the balance between highly specialised treatments, applicable to only a minority of patients and more widely applicable treatments favours the minority rather than the majority, the paper is now more balanced and gives a useful insight into a number (but not all) of novel treatment approaches.
Author Response
Comment: The authors have addressed many of the concerns in my previous review and, whilst personally, I think the balance between highly specialised treatments, applicable to only a minority of patients and more widely applicable treatments favours the minority rather than the majority, the paper is now more balanced and gives a useful insight into a number (but not all) of novel treatment approaches.
Reply: Many thanks for your kind and encouraging comments.